

# *Licinophilus depressus* n. gen., sp. n. (Eugregarinida: Stenophoridae) from *Licinus depressus* (Coleoptera: Carabidae)

Viktoriia Lazurska[1] and  Viktor V. Brygadyrenko[2]

[1] Unaffiliated, Dnipro, Ukraine

[2] Biodiversity and Ecology, Oles Honchar Dnipro National University, Dnipro, Ukraine

## ABSTRACT

The paper focuses on a new species of gregarines—*Licinophilus depressus* n. gen. sp. n., which was found in *Licinus depressus* (Paykull, 1790) (Coleoptera, Carabidae). The individuals of *L. depressus* ground beetle were collected on the bank of the Dnipro River, Ukraine and examined for the presence of gregarines in the guts. In the parasites discovered in the midguts, we measured the morphological parameters and their ratios, analyzing how they change in relation to the age of the specimens. According to our measurements, the species turned out to be incomparable to any of the ones described in the scientific literature. Because the gregarine specimens formed syzygies at late stages, were observed to have monogenous life cycle and a septum between the protomerite and deutomerite at all stages of development, and also rudimentary epimerite, they were classified to the Septatorina suborder, Stenophoricae superfamily, Stenophoridae family, and distinguished into a new genus due to the fluctuating septum between the protomerite and deutomerite.With age, the width of the segments does not change, while the gregarine increases in size due to significant elongation of the deutomerite. The most varying parameters of this species of gregarines were the deutomerite length (DL), the deutomerite length axis (DLA), and the total length of gregarine (TL) for which the coefficient of variation exceeded 30%. All the studied morphometric indices, except for the ratio of the deutomerite length to the deutomerite width (DL/DW), exhibited weak association with the age and sizes of gregarines. The indices such as the ratios of the epimerite length to the length of the protomerite-epimerite complex (EL/PECL), the protomerite length to the protomerite-epimerite complex (PL/PECL), the epimerite length to the epimerite width (EL/EW), the protomerite length to the protomerite width (PL/PW), the protomerite width to the septum width (PW/SW), the deutomerite width to the protomerite width (DW/PW), the nucleus length to nucleus width (NL/NW), and also DLA remained constant in this gregarine during growth, and therefore they can be used as constant diagnostic features.

Corresponding author
Viktor V. Brygadyrenko,
brigad@ua.fm

## INTRODUCTION

The phylum Apicomplexa comprises single-cell parasites and contains significant pathogenic organisms such as *Plasmodium malariae* and *Toxoplasma*. Gregarines of this phylum parasitize a broad range of invertebrates, both in water and land (*Mahé et al., 2017*). Gregarines are considered one of the earliest branches of Apicomplexa (*Leander, Clopton & Keeling, 2003*), which makes them great models for studying the evolution of this type (*Wakeman & Leander, 2012*). Due to a number of specific traits of gregarines, numerous authors have been actively conducting relevant phylogenetic and taxonomic studies of this subclass (*Janouškovec at al., 2019*; *Mathur et al., 2019b*; *Iritani et al., 2021*). Their functioning and evolution can shed light on the life of other Apicomplexa protozoans due to the unique structure and their adaptation strategies (*Wakeman & Leander, 2012*; *Boisard & Florent, 2020*). Gregarines are also considered a good example of long co-evolution of host and parasite (*Valigurová, 2012*). As of now, 1,770 species of gregarines have been described (*Portman & Slapeta, 2014*; *Votýpka et al., 2017*). Studies have demonstrated that gregarines comprise the main group of protists in the tropical forests (*Mahé et al., 2017*). They are specific to hosts and parasitize almost all invertebrates, including insects (*Levine, 1988*). Most gregarines can developed only in a limited number of species of invertebrates or even in a particular development stage of the host (*Clopton, Janovy & Percival, 1992*). Only 0.33% of known species of insects have been analyzed for gregarines; the actual number of gregarine species can account for millions (*Levine, 1988*; *Clopton, 2002*). The recent studies of soils and marine environments demonstrated a wide distribution and diversity of gregarines (*Boisard, 2021*). Thus, the diversity of gregarines is significantly understudied, which hinders the process of researching their effects on the hosts.

Gregarines enter the host organism when the host ingests sporocysts and live in its intestines, Malpighian tubule system, or the tissues (*Gigliolli, Julio & Conte, 2016*). Released sporozoites attach to the host's cells by an epimerite, which at later development stages can remain or disappear (*Valigurová et al., 2007*; *Valigurová, Michalková & Koudela, 2009*). Further, gregarine development is extracellular. Gregarines come in a variety of sizes and forms, and are characterized of fast development: some species are able to grow from less than 1 micrometer to 1–2 milimeters in about two weeks (*Desportes & Schrével, 2013*). During growth, gregarines consume nutrients from the host body, although whether this has a significant impact on the host is yet to be determined (*Valigurová & Florent, 2021*; *Parhomenko & Brygadyrenko, 2023*; *Lazurska & Brygadyrenko, 2024*). Mature gamonts form syzygies, which later form new gametocysts. The presence of late or early syzygies is one of the main diagnostic criteria for gregarines (*Clopton, 2002*). Other criteria include morphometric parameters, duration of the life cycle phases, type of syzygy, peculiarities of their location inside the host, and taxonomical composition of hosts (*Boisard, 2021*).

The symbiosis between gregarious insects and their hosts remains a subject of much debate: views on it vary from parasitism to mutualism (*Rueckert, Betts & Tsaousis, 2019*; *Barber, Friedrichs & Müller, 2024*). Mostly, gregarines are commensals, but in multiple situations their presence can affect the organisms of invertebrates, their hosts. For example, they can hinder growth, cause arrested development, and decrease the lifespan of its

host (*Canales-Lazcano, Contreras-Garduño & Córdoba-Aguilar, 2005*). Combined with other factors, such as infection with microsporidia (*Fellous & Koella, 2010*) or impact of pesticides (*Wolz et al., 2022*), gregarines exacerbate the host's condition. Moreover, an excessive number of gregarines in the intestines leads to problems with food metabolism and can result in the death of the affected hosts (*Mita et al., 2012*). At the same time, gregarines help the host to survive in the circumstances of non-optimal diets, and their numbers correlate with its longer life in damselflies *Enallagma boreale* (Selys, 1875) (Odonata: Coenagrionidae) and *Victorwithius similis* (Beier, 1959) (Arachnida: Pseudoscorpiones) (*Hecker, Forbes & Leonard, 2002*; *Bollatti & Ceballos, 2014*). The mechanisms of their influence on the host are ambiguous and require further research (*Parhomenko et al., 2023*).

Ground beetles are one of the most numerous families of coleopterans (*Erwin, Micheli & Chaboo, 2015*). In Ukraine, 752 species have been recorded (*Puchkov, 2018*), including 281 in Dnipropetrovsk Oblast (*Brygadyrenko, 2003*). They serve as bioindicators for many biotic and abiotic factors, including environmental pollution (*Koivula, 2011*; *Kotze et al., 2011*; *Cividanes, Cividanes & Ferraudo, 2017*). Some of the factors influencing the populations of ground beetles are their pathogens and parasites (*Lövei & Sunderland, 1996*).

Gregarines of different insects are closely associated with their hosts. Nonetheless, the fauna of gregarines of ground beetles has been described fragmentarily, the common species being those studied the most (*Desportes & Schrével, 2013*; *Kobeza & Pakhomov, 2019*). Of the Palearctic tribe Licinini, only *Licinus punctulatus* (Fabricius, 1792) has been a subject of such a research. This ground beetle was found to be typically parasitized by *Actinocephalus licini* Tuzet & Théodoridès, 1951b and *Ramicephalus licini* (Tuzet & Théodoridès, 1951b) Tuzet, Ormières & Théodoridès, 1968 (*Geus, 1969*; *Desportes & Schrével, 2013*). Meanwhile, *Licinus depressus* have not been mentioned in the scientific literature so far.

Thus, the purpose of this article is to describe a new species of gregarine we identified in the midgut of the imago *Licinus depressus*.

## MATERIALS AND METHODS

The imago specimens of *L. depressus* were collected between June and September of 2024 in the city of Dnipro (Central Ukraine). Field experiments were approved by the Research Council of the Oles Honchar Dnipro National University (project number: 0122U001225). To catch the beetles, we used Barber's pitfall traps (*Hohbein & Conwey, 2018*; *Litavský & Prokop, 2023*), which were placed in groups of 5 to 10 across several ecosystems in the territory of the bank of the Dnipro River: 1. 48°25′25.0″N, 35°04′46.1″E; 2. 48°25′30.5″N, 35° 04′48.3″E; 3. 48°25′31.9″N 35°04′47.3″E; 4. 48°25′31.0″N 35°04′44.9″E. The transparent plastic cups, each with a 500 mL capacity and five small holes at the bottom to allow rainwater to drain away, were installed in the soil and were regularly checked for the presence of insects (*Willand & McCravy, 2018*). Upon discovery, the invertebrates were placed to the general container and examined for the presence of gregarines in the subsequent 24–48 h.

To diagnoze gregarines in the beetles, their intestines were removed and transferred to microscope slide in physiological solution and cut into 14–16 transversal sections at equal

distance one from another (*Brygadyrenko & Reshetniak, 2016*). Under a light microscope, the intestines were analyzed for the presence of gregarines.

Of the 34 examined individuals of *L. depressus* ground beetles, gregarines were found in 16 (the prevalence measuring 47%), with the average load of 14.8 specimens per beetle. All gregarines found belonged to one species (Fig. 1).

For further morphometric analysis, we took photos using a camera with 5 megapixel resolution. The morphometric parameters were measured using the TpsDig 2.17 software (2013, Rohlf FJ, Ecology & Evolution, SONY at Stony Brook). We used 30 single gamonts for the measurements, and analyzed over 150 gregarines at various development stages to diagnoze the species. The parameters included the standard measurements for gregarines and the additional mesurements proposed by *Clopton & Nolte (2002)*. Further analysis of data was conducted using the Statistica software (version 8, StatSoft, USA). To analyze the variability of gregarines, mean values, standard deviations, minimal and maximal values of the characteristics, and coefficient of variation of the sampling were calculated (Tables 1 and 2). A regression analysis of the dependence of morphometric characteristics and indices on the length of gregarines was performed (Figs. 2–5).

The electronic version of this article in Portable Document Format (PDF) will represent a published work according to the International Commission on Zoological Nomenclature (ICZN), and hence the new names contained in the electronic version are effectively published under that Code from the electronic edition alone. This published work and the nomenclatural acts it contains have been registered in ZooBank, the online registration system for the ICZN. The ZooBank LSIDs (Life Science Identifiers) can be resolved and the associated information viewed through any standard web browser by appending the LSID to the prefix http://zoobank.org/. The LSID for this publication is: LSIDurn:lsid:zoobank.org:pub:C7061E95-5071-47B5-B4BE-177ACB9B6684. The online version of this work is archived and available from the following digital repositories: PeerJ, PubMed Central SCIE and CLOCKSS.

## RESULTS

### Generic diagnosis

Order Eugregarinida Leger, 1892; Suborder Septatina Lankester, 1885; Superfamily Stenophoricae Levine, 1984; Family Stenophoridae, Léger & Duboscq, 1904; Genus *Licinophilus* Lazurska & Brygadyrenko, 2025.

The gregarines in the midgut of *L. depressus* were in the form of trophozoites or gamonts. The youngest trophozoites, measuring around 50 μm, remained a complex epimerite with appendages, which were reduced in the trophozoites at later developedment stages.

The reduced epimerite constituted a delta-like structure, covering the protomerite as a crescent or a semi-circle in trophozoites and was elongated into a separate segment in gamonts. At different development stages of the parasite, some individuals had a distinct endoplasmatic septum with protomerite or completely fused with it. The protomerite had varying oval shape. The septum between the protomerite and deutomerite was distinctly notable, sometimes slightly narrowed, or of a consistent width. The deutomerite was thin

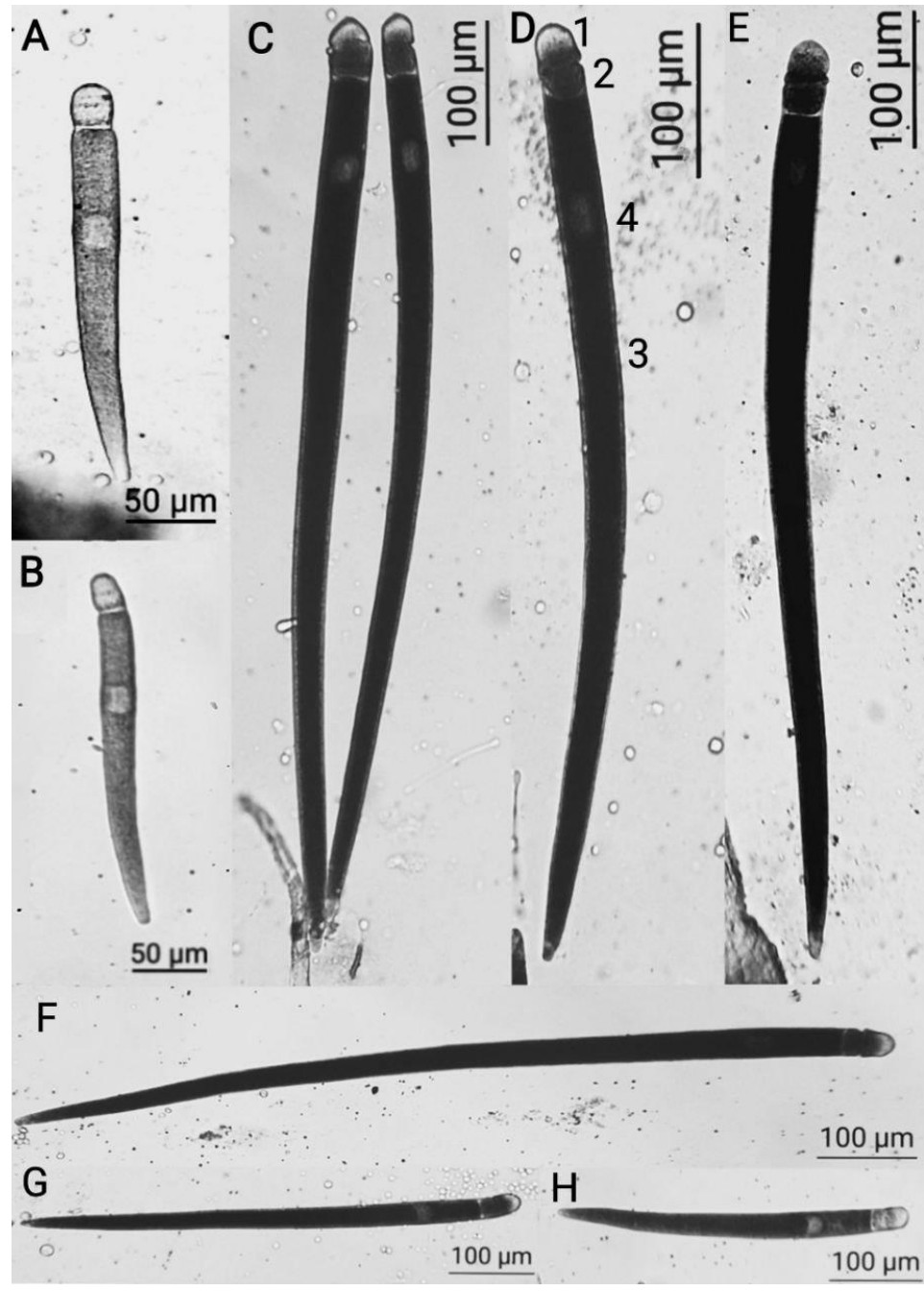

**Figure 1** *Licinophilus depressus* **sp . n.** (A–B, H) trophozoites with reduced epimerite, the septum between the protomerite and epimerite is not distinct; (C–G) gamonts, with notable septum between the protomerite and epimerite; 1, epimerite; 2, protomerite; 3, deutomerite; 4, nucleus.

**Table 1** Morphometric characteristics of individual gregarines, mean values, standard deviations (SD), minimal and maximal values of the characteristics, and coefficient of variation of the sampling (CV, expressed in %; $N = 30$).

| Characteristic | PECL | EL | EW | PL | PW | SW | DL | DW | DLA | TL | NL | NW | NDS |
|---|---|---|---|---|---|---|---|---|---|---|---|---|---|
| Mean | 51.4 | 31.0 | 29.5 | 23.1 | 30.6 | 30.3 | 623.8 | 33.9 | 65.7 | 673.7 | 23.2 | 19.9 | 57.3 |
| SD | 11.7 | 8.4 | 6.1 | 5.3 | 5.2 | 5.5 | 209.2 | 5.1 | 21.6 | 218.6 | 5.0 | 5.0 | 18.6 |
| Min | 32.8 | 19.4 | 18.7 | 11.9 | 21.6 | 23.1 | 334.9 | 25.5 | 12.0 | 374.1 | 11.9 | 11.8 | 23.1 |
| Max | 93.7 | 61.4 | 52.0 | 35.0 | 45.3 | 45.8 | 1,102.6 | 48.6 | 101.2 | 1,169.1 | 31.7 | 33.6 | 88.8 |
| CV, % | 22.8 | 27.0 | 20.7 | 22.8 | 17.0 | 18.0 | 33.5 | 15.0 | 32.9 | 32.5 | 21.6 | 24.9 | 32.5 |

Notes.

PECL, the length of the protomerite–epimerite complex; EL, the epimerite length; EW, the epimerite width; PL, the protomerite length; PW, the protomerite width; SW, the width of the protomerite-deutomerite septum; DL, the maximum length of the deutomerite; DW, the protomerite width; DLA, the distance from the protomerite-deutomerite septum to the deutomerite axis of maximum width; TL, the total length; NL, the nucleus length; NW, the nucleus width; NDS, the distance from the nucleus to the protomerite-deutomerite septum.

**Table 2** Morphometric indices (expressed in relative values) of the individual specimens of gregarines, mean, standard deviation, minimal and maximal value of the characteristic, and the coefficient of variation of the sampling (CV, expressed in %; $N = 30$).

| Index | EL/PECL | PL/PECL | EL/EW | PL/PW | PW/SW | DL/DW | DL/PL | DW/PW | TL/EL | TL/PL | TL/DL | NL/NW |
|---|---|---|---|---|---|---|---|---|---|---|---|---|
| Mean | 0.601 | 0.450 | 1.047 | 0.758 | 1.019 | 18.1 | 27.6 | 1.114 | 21.8 | 29.9 | 1.084 | 1.250 |
| SD | 0.073 | 0.078 | 0.155 | 0.179 | 0.062 | 4.3 | 8.0 | 0.059 | 4.2 | 8.3 | 0.018 | 0.454 |
| Min | 0.419 | 0.320 | 0.734 | 0.421 | 0.926 | 11.0 | 13.2 | 0.951 | 14.3 | 14.8 | 1.058 | 0.467 |
| Max | 0.749 | 0.611 | 1.325 | 1.368 | 1.163 | 26.2 | 41.5 | 1.212 | 30.5 | 44.1 | 1.123 | 2.507 |
| CV, % | 12.2 | 17.4 | 14.8 | 23.6 | 6.1 | 23.6 | 28.8 | 5.3 | 19.4 | 27.6 | 1.7 | 36.3 |

Notes.

EL/PECL, the ratio of the epimerite length to the length of the protomerite–epimerite complex; PL/PECL, the ratio of the protomerite length to length of the protomerite–epimerite complex; EL/EW, the ratio of the epimerite length to the epimerite width; PL/PW, the ratio of the protomerite length to the protomerite width; PW/SW, the protomerite width to the width of the protomerite-deutomerite septum; DL/DW, the ratio of the deutomerite length to the maximum deutomerite width; DL/PL, the ratio of the deutomerite length to the protomerite length; DW/PW, the ratio of the maximum deutomerite width to the protomerite width; TL/EL, the ratio of the total length to the epimerite length; TL/PL, the ratio of the total length to the protomerite length; TL/DL, the ratio of the total length to the deutomerite length; NL/NW, the nucleus length to the nucleus width.

and elongated, slightly expanding around the location of nucleus, and narrowing and sharpening towards the end. The nucleus was located in the upper half of deutomerite, ranging in shape from rounded and oval to sicle-shaped.

## Morphometric parameters

The coefficient of variation indicated the stability of morphometric parameters and indices (Tables 1 and 2). Of the 13 morphometric parameters, the least variable (minimal value of the coefficient of variation) was the deutomerite width. With age, the deutomerite diameter in this species of gregarines remained almost unchanged. Less constant, although the most consistent among the rest characteristics, were the protomerite, and the septum width. A relatively stable characteristic was the epimerite width. The least variable parameter in this species of gregarine was observed to be the maximal deutomerite length, the distance from the septum to the axis of maximal deutomerite width, and the total length of gregarine, for which the coefficient of variation exceeded 30%.

The most steady morphometric parameters (Table 2) were the total length of gregarine to the deutomerite length (TL/DL), the deutomerite width to the protomerite width (DW/PW), the protomerite width to the septum width (PW/SW). Significantly higher individual variability was seen in the morphometric indices such as the ratios of

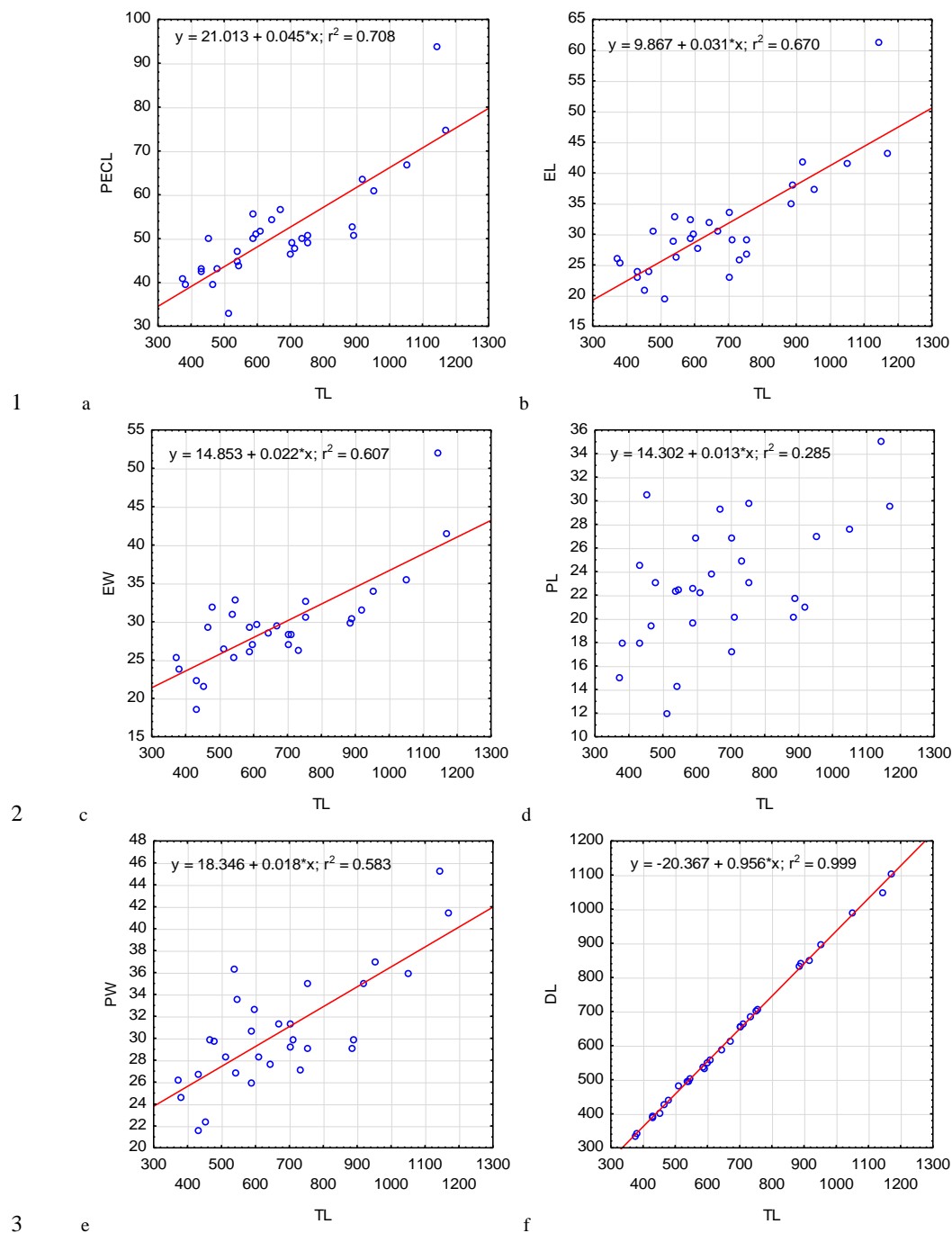

**Figure 2  Changes in the length of protomerite–epimerite complex.** PECL (A); the length of the epimerite, EL (B); the width of the epimerite, EW (C); the length of the protomerite, PL (D), the width of the protomerite, PW (E); the maximum length of the deutomerite, DL (F), depending on the total length, TL (on the abscissa axis): values of all the characteristics on the abscissa axis and the ordinate axis are presented in micrometers; $n = 30$; in the equation of linear regression, x- length of gregarine in micrometers, y- characteristic indicated on the ordinate axis (in micrometers); $r^2$- coefficient of determination that varies 0 to 1.

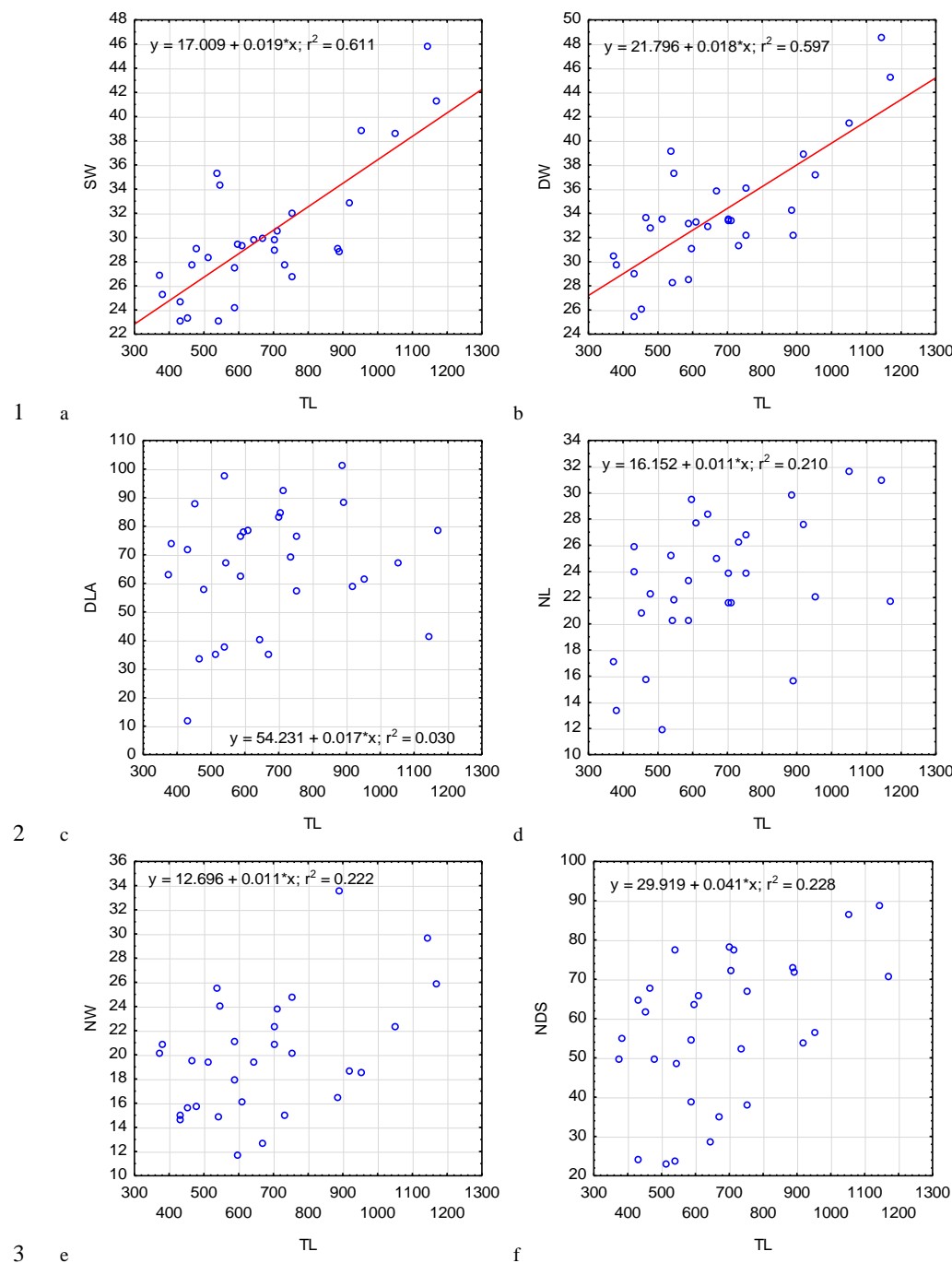

**Figure 3** Changes in the width of the protomerite-deutomerite septum SW (A), width of the protomerite DW (B), distance from the protomerite-deutomerite septum to the deutomerite axis of maximum width DLA (C), length of the nucleus NL (D), width of the nucleus NW (E), distance from the nucleus to the protomerite-deutomerite septum NDS (F) depending on the total length TL (along the abscissa axis): for explanation see **Fig. 2**.

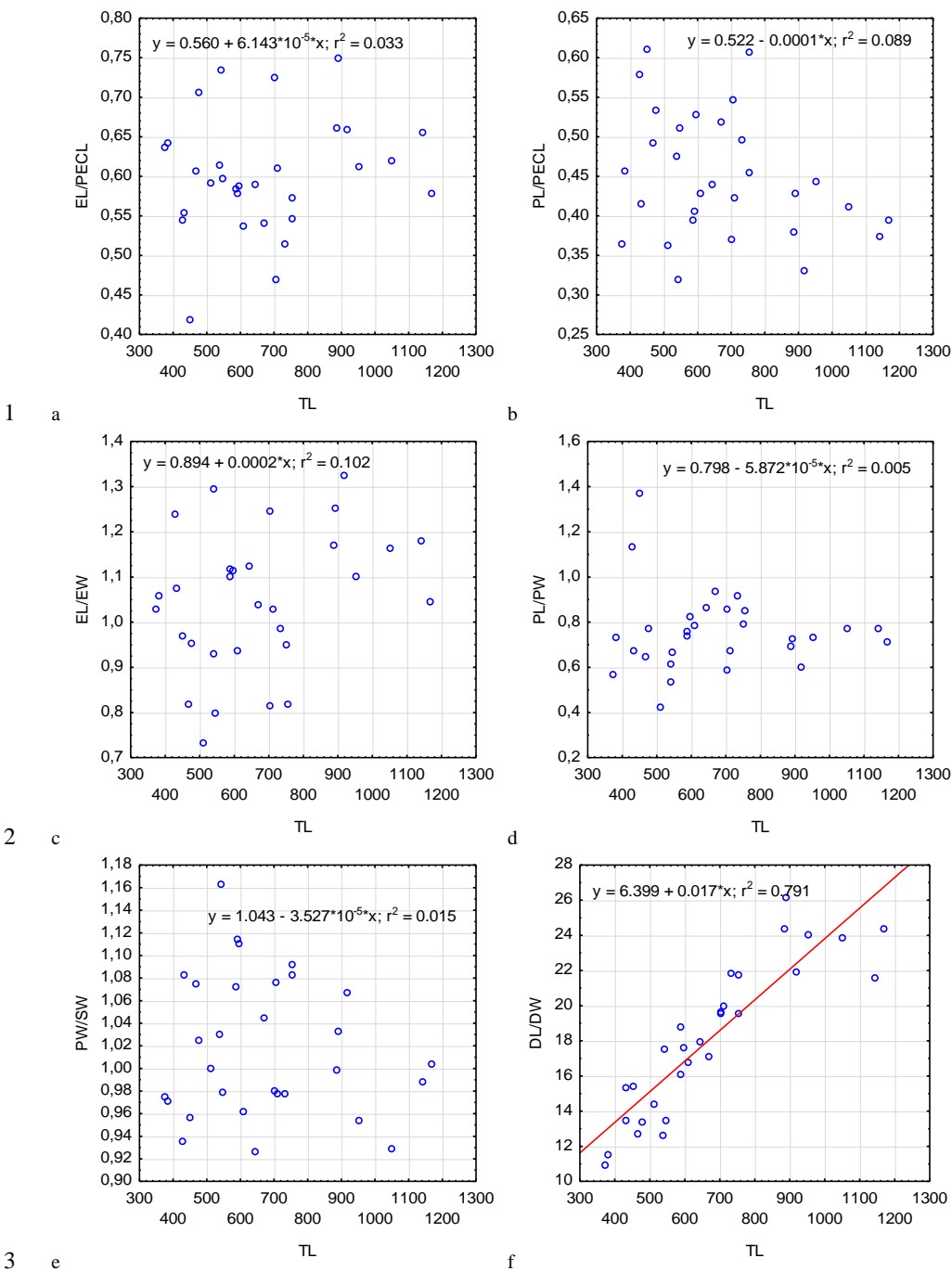

**Figure 4** Changes in the ratio of the epimerite length to the length of the protomerite–epimerite complex, EL/PECL (A); ratio of the protomerite length to the length of the protomerite–epimerite complex, PL/PECL (B), the ratio of the epimerite length to the epimerite width, EL/EW (C); the ratio of the protomerite length to the protomerite width, PL/PW (D); the protomerite width to the width of the protomerite-deutomerite septum, PW/SW (E); the ratio of the deutomerite length to the maximum width of the deutomerite, DL/DW (F), depending on the total length TL (on the abscissa axis): see explanations in Fig. 2.

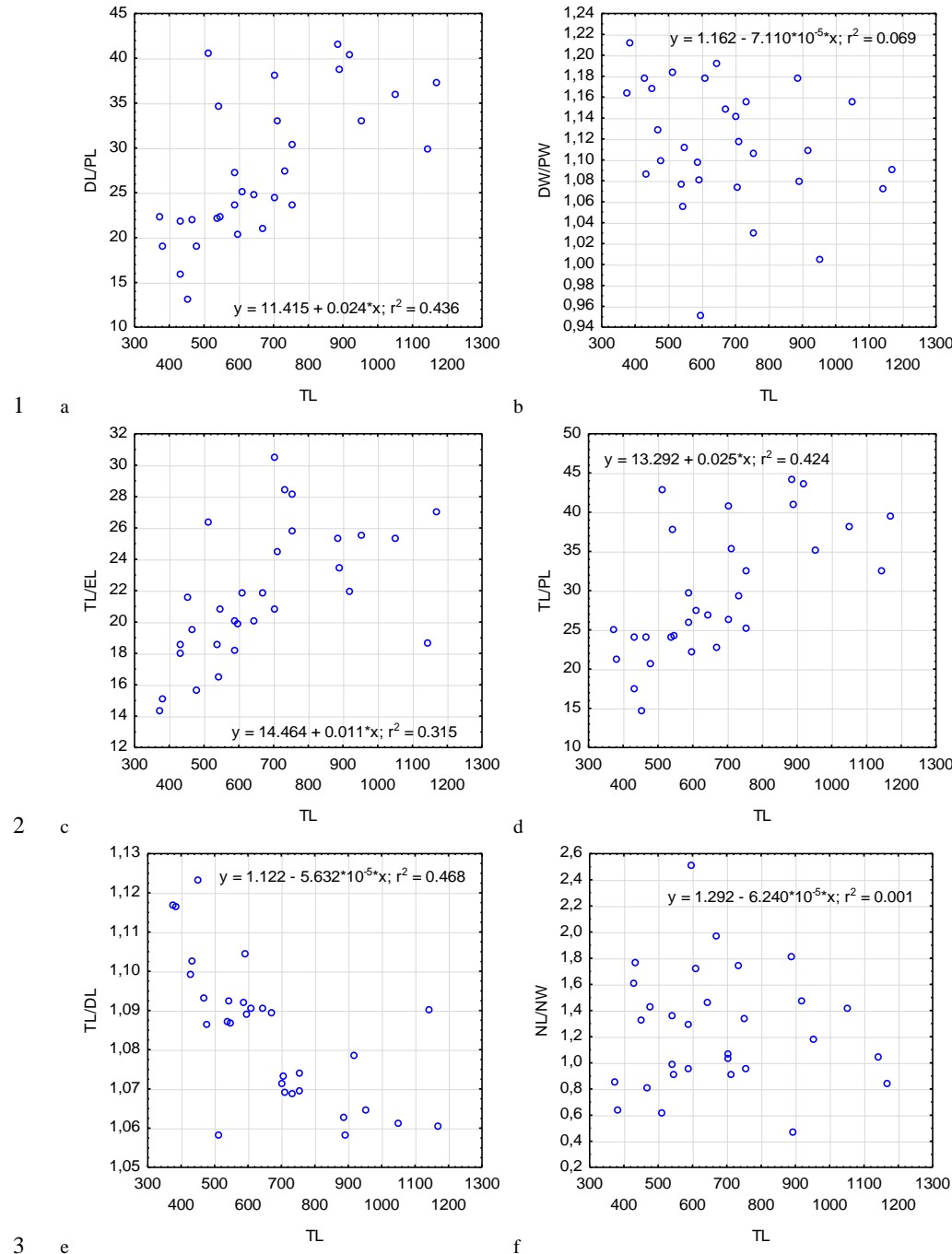

**Figure 5** Changes in the ratio of the deutomerite length to the protomerite length, DL/PL (A); the ratio of the maximum deutomerite width to the protomerite width, DW/PW (B); the ratio of the total length to the epimerite length, TL/EL (C); the ratio of the total length to the protomerite length, TL/PL (D); the ratio of the total length to the deutomerite length, TL/DL (E); the nucleus length to the nucleus width, NL/NW (F), depending on the total length TL (on the abscissa axis): see explanations in Fig. 2.

the epimerite length to the length of the protomerite-epimerite complex (EL/PECL), the epimerite length to the epimerite width (EL/EW), the protomerite length to the protomerite-epimerite complex (PL/PECL), the protomerite length to the protomerite width (PL/PW), the ratio of the deutomerite length to the deutomerite width (DL/DW), the total length of gregarine to the protomerite length (TL/PL), the ratio of the deutomerite length to the protomerite length (DL/PL), and the nucleus length to nucleus width (NL/NW).

Across the gregarines specimens, some characteristics and indices in the gregarines largely varied, while other were almost the same (Figs. 2–5). The closest correlation with the total length of gregarines of this species was observed for the deutomerite length, DL ($r^2 = 0.999$, Fig. 2F). Further, the parameters are presented in descending order of the coefficients of determination: DL/DW ($r^2 = 0.791$, Fig. 4F), PECL ($r^2 = 0.708$, Fig. 2A), EL ($r^2 = 0.670$, Fig. 2B), SW ($r^2 = 0.611$, Fig. 3A), EW ($r^2 = 0.607$, Fig. 2C), DW ($r^2 = 0.597$, Fig. 3B), PW ($r^2 = 0.583$, Fig. 2E). For the other morphometric characteristics we measured, the coefficient of determination ($r^2$) was less than 0.5, particularly, PL ($r^2 = 0.285$, Fig. 2D), deutomerite length axis (DLA) ($r^2 = 0.030$, Fig. 3C), NL ($r^2 = 0.210$, Fig. 3D), NW ($r^2 = 0.$ , Fig. 3E), and NDS ($r^2 = 0.228$, Fig. 3F). All the studied morphometric indices, except for DL/DW, were weakly associated with the size of gregarines. For 11 of 12 indices, the $r^2$ value was below 0.5: NL/NW ($r^2 = 0.001$, Fig. 5F), PL/PW ($r^2 = 0.005$, Fig. 4D), PW/SW ($r^2 = 0.015$, Fig. 4E), EL/ PECL ($r^2 = 0.033$, Fig. 4A), DW/PW ($r^2 = 0.069$, Fig. 5B), PL/PECL ($r^2 = 0.089$, Fig. 4B), EL/EW ($r^2 = 0.102$, Fig. 4C), TL/EL ($r^2 = 0.315$, Fig. 5C), TL/PL ($r^2 = 0.424$, Fig. 5D), DL/PL ($r^2 = 0.436$, Fig. 5A), and TL/DL ($r^2 = 0.468$, Fig. 5E).

The characteristics that remained almost unchanged during growth of the gregarines (*i.e.,* with increase in their general length) were those whose coefficient 'a' in the equation of linear regression was closest to zero, and whose trend line on the dispersion diagram was located almost horizontally. For example, these were EL/PECL, PL/PECL, EL/EW, PL/PW, PW/SW, DW/PW, NL/NW, and DLA. These parameters remained constant in this gregarine over the process of growth, and therefore can be used as consistent diagnostic traits.

## DISCUSSION

The species we studied belongs to the Apicomplexa phylum, Gregarinasina subclass, Eugregarinorida order. Due to the distinct separation of the protomerite and deutomerite by the endoplasmatic septum, we placed it in the Septatorina suborder. According to monogenic life cycle and late formation of syzygies, the species was classified to the Stenophoricae superfamily. According to the diagnostic criteria presented by *Clopton (2002)*, the gregarine was assigned to the Stenophoridae family. The criteria included the presence of a septum between the protomerite and deutomerite at all the stages of development (sporozoite to trophozoite) and a reduced, rudimentary, or concave epimerite. *Licinophilus depressus* is the first gregarine of the Stenophoridae family that was discovered in ground beetles. The Stenophoridae family contains three genera: *Hyalosporina* Chakravarty, 1935, *Fonsecaia* Pinto, 1918, and *Stenophora* Labbé, 1899 (*Clopton, 2002*).

None of these corresponded to the species we described: prior to this, all gregarines of the Stenophoridae family were found in millepedes. The rudimentary epimerite of those species had no clear septum separating it from the protomerite, and remained as a small formation. In our case, the species was found in the representative of the family of ground beetles, and the reduced epimerite was a small "crescent" that covered the protomerite, divided by a septum at certain stages of development or completely fused with the protomerite. Therefore, we propose designating the new genus *Licinophilus*, to which the species we described will be classified under the name *Licinophilus depressus*.

Most of the research on gregarines of ground beetles was conducted for common carabid species (*Sienkiewicz & Lipa, 2009*; *Desportes & Schrével, 2013*). The gregarines found in them usually belong to related families and genera. According to the literature sources, *Licinus punctulatus* (Fabricius, 1792) was observed to harbor *Actinocephalus licini* Tuzet & Théodoridès, 1951b and *Ramicephalus licini* (Tuzet & Théodoridès, 1951b) Tuzet, Ormières & Théodoridès, 1968 (*Geus, 1969*; *Desportes & Schrével, 2013*). The species we encountered is distinct from the *Actinocephalus* genus by the absence of a developed epimerite with appendages in gamonts and trophozoites, and also the absence of a neck-like septum. Some species of the *Actinocephalus* genus (*Actinocephalus acutispora* Leger, 1892 and *Actinosephalus permagnus* Wellmer, 1911) have similar morphological parameters, but do not have an expressed endoplasmatic septum of the protomerite (*Geus, 1969*). From the *Ramicephalus* genus, the species is different by the absence of a permanent epimerite with numerous appendages, and also by a larger size (*Clopton, 2002*). The species we described appears similar to the *Clitellocephalus* genus, although does not form early syzygies, and the epimerite is able to fuse with the protomerite, whereas in the *Clitellocephalus* genus the epimerite remains permanently (*Clopton & Nolte, 2002*). Some scientific works have reported the presence of gregarines of the Stylocephalidae family in ground beetles, although, despite the visual similarity, gregarines of the Stylocephalidae family have an elongated epimerite that is located on the neck-like narrowing, which we did not observe (*Clopton, 2002*). Many gregarines found in ground beetles belong to the Gregarinicae superfamily. However, due to the early formations of syzygies that is characteristic of this family we cannot classify the described species to it. A similar phenomenon of fluctuating septum between the epimerite and protomerite was observed in Amoebogregarina (*Kula & Clopton, 1999*), though while the epimerite in Amoebogregarina integrates into the protomerite when the gamont becomes mature, in our case the septum more often emerged at late stages of the development, and the mechanisms of integration of the epimerite remain unknown.

The analysis of 30 single specimens of *Licinophilus depressus* demonstrated a substantial variability in some parameters and indices. The most variable parameters in this species of gregarines were the maximum deutomerite length (DL), distance from the septum to the deutomerite width axis (DLA), and total length of gregarine (TL), for which the coefficient of variation was over 30%. These parameters were the most fluctuating among the gregarines of different size. The least variation was observed in EW, PW, SW, and DW, the coefficient of variation not exceeding 21%. This demonstrated that during its growth, *Licinophilus depressus* elongates mostly due to the deutomerite, while the width of the segments remain

more stable. A similar tendency was described for the development of *C. ophoni* (Tuzet & Ormieres, 1956) (*Clopton, 2002*; *Clopton & Nolte, 2002*) and *A. permagnus* (*Geus, 1969*).

The results of our study turned out to be unexpected. The available literature (*Sienkiewicz & Lipa, 2009*; *Desportes & Schrével, 2013*) contains no reports that gregarines of the Stenophoridae family parasitize ground beetles, although it is important to consider that not all species of the *Licinus* genus were examined for gregarines: of the 29 known species, there is only data regarding *L. punctulatus*. The Licinini tribe requires further research for a better understanding of how gregarines adapt to these different host species and for a better knowledge of their co-evolution. We designate *Licinophilus depressus* into a separate genus *Licinophilus* and consider it species and genus that are new to science.

## CONCLUSIONS

This study describes a new species and genus of gregarines –*Licinophilus depressus* and *Licinophilus*, found in the ground beetle *Licinus depressus*. Our findings provide new insights into the biodiversity and contribute to the yet understudied scientific area that is gregarines in insects, particularly, carabids. Considering the importance of gregarines as models for studies in evolution and medicine and the significance of ground beetles in agriculture, our research can contribute to new developments in the these spheres. Moreover, carabids are indicators of the environmental changes (*Niemelä & Kotze, 2009*), and thus this new discovery is another argument for preserving the delicate balance of our ecosystem. We strongly recommend continuing research of insect parasites, gregarines, in particular, to achieve a better understanding of their relationships with hosts and potential impacts on host populations.

### Funding
The authors received no funding for this work.

### Competing Interests
Viktor V. Brygadyrenko is an Academic Editor for PeerJ.

### Author Contributions
- Viktoriia Lazurska conceived and designed the experiments, performed the experiments, analyzed the data, prepared figures and/or tables, authored or reviewed drafts of the article, and approved the final draft.
- Viktor V. Brygadyrenko conceived and designed the experiments, analyzed the data, prepared figures and/or tables, authored or reviewed drafts of the article, and approved the final draft.

### Field Study Permissions
The following information was supplied relating to field study approvals (i.e., approving body and any reference numbers):

Field experiments were approved by the Research Council of the Oles Honchar Dnipro National University (project number: 0122U001225).

## Data Availability

The raw measurements are available in the Supplementary File.

## New Species Registration

The following information was supplied regarding the registration of a newly described species:

Publication LSID: urn:lsid:zoobank.org:pub:C7061E95-5071-47B5-B4BE-177ACB9B6684

Licinophilus n. gen. LSID: urn:lsid:zoobank.org:act:E31A0D32-B638-4B7B-8284-5F69611550BB

Licinophilus depressus n. sp. LSID: urn:lsid:zoobank.org:act:A6307690-6A04-45CF-8DCB-48BF9B71ED41.

## Supplemental Information

Supplemental information for this article can be found online at http://dx.doi.org/10.7717/peerj.20099#supplemental-information.

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
