# Peer review of "Licinophilus depressus n. gen., sp. n. (Eugregarinida: Stenophoridae) from Licinus depressus (Coleoptera: Carabidae)"

_PeerJ, doi:10.7717/peerj.20099_

## Round 0.1 · original submission · Minor Revisions

Reviewer 1 ·

Basic reporting

The authors have found a new gregarine species in a beetle, which they describe morphologically in detail. Overall, they did a great job in describing the different parameters, and it is exciting that they found, most likely, a new species. We still know very little about gregarines, so work in this direction is needed. However, the introduction could be more focused on the own work. Also, I was surprised to see no data for gregarine stages in the larvae of the beetles. Overall, the grammar needs some polishing. I pointed out some instances, but there are more mistakes.

Abstract:
First sentence: Please add that the host is a Coleoptera, Carabidae.
Second sentence: add “(Ukraine)” after “Dnipro River”.
Maybe add in the abstract where the parasites are found. I assume in the guts?
Line 28: “With age, the width of the segments remained (change, instead of remains) unchanged…”

Introduction:
Line 60: by an epimerite (add “an” before “epimerite”)

Line 61: “The attachment strategy of gregarines is similar to that of cryptosporidia.” This sentence can be deleted if nothing more is said about the attachment strategy.

Line 63: change “and also are characteristic of fast development” to “and are characterized by fast development”.

Line 72-73: Please rephrase this sentence and use correct grammar.

Line 72, you say “invertebrates”, in line 75 “insects” – maybe stick to “invertebrates” here.

Lines 74 and 79: can affect the hosts (replace “organism” by “host”)

Line 80-81: Probably, this is specific for a certain host species and its gregarine, and not a general phenomenon. So it should be specified where this phenomenon was found.

Lines 84-95: There is some redundancy; this paragraph could be drastically shortened and the information moved to the end of the first paragraph. In that way, one gets an idea early on, which research gap this study aims to fill.

Line 76-105: This paragraph could also be shortened, and a stronger focus given on what is studied here. Not all information on ground beetles is needed here, as this was no comparative work, but you worked just on one species. Instead, please end the introduction with the aim of the current study.

Line 107: What do you mean by “mass species”? - also later in line 225.

Line 110: Was found to be typically (add “be”)

Methods/experimental design:
Were only adult beetles analyzed? What about the larvae? Does this gregarine species only colonize adults, or in which stages are they found in the larvae?

Did you find differences between male and female host beetles in colonization loads?

Line 133: check spelling “software”

Was this beetle species only colonized by one gregarine species?

How many beetles did you dissect in total? This comes in the results bot should also be mentioned in the method section.

Line 136: “Further analysis of data” – Please specify: which exact statistical analyses were performed and on which data?

Results: Please make clear throughout whether your description refers to the gregarine or its host (e.g., line 162 “at different developmental stages” – of the gregarine or the host?). Please check throughout.

Line 156: Please replace “infestation extensiveness” with “prevalence.”
And “average intensity” with “average load”

Line 161-.162: What is “more adult individuals” referring to: adult host or adult gregarine? And what is “more adult” compared to “adult”?

Line 163: “fused” (instead of “fuse”)?

It would be nice if you could refer in the generic description part to Figure 1. In Figure 1, one of the individuals should be labeled, so that readers unfamiliar with the exact morphology of gregarines know what/where the epimerite, protomerite, etc, are.

Line 185: across the specimens of different ages – do you mean the host here? How was their age determined?

Instead of repeating all values in the text that can also be seen in the Tables, you could just highlight values with low and high variation here and shorten the text, thereby drastically. The readers can read the exact values in the Tables.

Line 194: How did you determine the age of the gregarines?

Discussion:
Line 210: “According to monogenic life cycle” – add “the” before “monogenic” – but how did you determine that the life cycle is monogenic?

Line 214: “at all the stages of development” – please specify: which developmental stages do you mean? Not all gregarine developmental stages are structured like this.
Line 215-216: Do you mean you have now described the first gregarine of the Stenophoridae family in a ground beetle? Then clarify this and delete the reference to Desportes and Schrevel in the end, because they did not describe this!

The next sentence about the three genera needs a reference instead.

Line 219: Maybe start a new sentence when you talk about the rudimentary epimerite – or was that found in the gregarines from the millipedes? This is not clear.

Line 235: absence of a permanent epimerite (add “a”)

Line 242: check spelling of “beetles”

Line 244: Delete the Clopton reference at the end of the statement, as this sentence refers to your own finding.

Line 245: A similar phenomenon of fluctuating septum between the epimerite and protomerite was observed in Amoebogregarina – please add a reference for this statement here.

Line 248: Please start a new sentence for the last statement and change the wording: “The mechanisms of integration of the epimerite remain unknown (Kula & Clopton, 1999).”

Line 254: Again, whose age and how did you determine this? Could differences also depend on host size or sex?

Line 264: Maybe rather “adapt to these different host species”?

Last sentence: “Thus” does not fit; this does not connect with the sentence before.

Line 272: “underscores the miscellaneous tapestry of life on Earth.” – Please delete.

Line 274: I think this also holds for many other beetle families, not only carabids.

Line 277: “carabids are indicators of environmental changes” – this statement needs a reference – why is that so?

In the introduction, you wrote a lot about the potential impacts of gregarines on the host. So maybe you could also come back to that in the conclusion and write that more studies are needed also on the biology of this host-gregarine relationship to understand their roles/functions.


Figures and Tables:
Please add in the legends of Tables 1 and 2, on how many individuals the means are based on.

Legend for Fig. 5: Something seems to be missing in the legend.

Legend of Fig. 3: The last sentence is not clear.

In Fig. 3, the values for DLA do not seem to be statistically linearly correlated, at least the R2 is very low. Did you always test whether the linear correlation is indeed significant? Please only add the red lines and linear regression function in those figures, where the data are indeed significantly linearly correlated.

The same holds particularly for Fig. 4, in which sometimes low R2 values are given. The red line would imply a correlation that may not exist at all, so please check all figures.

Experimental design

-

Validity of the findings

-

Reviewer 2 ·

Basic reporting

The authors of the present manuscript describe a new species of gregarine (Licinophilus depressus, Eugregarinida, Stenophoridae), found in the beetle Licinus depressus. According to the accepted practice, they have presented the necessary data and morphometric indices and proportions/ratios of the body parts of the parasite, showing that it differs morphologically and structurally from other closely related species. The data are registered in ZooBank and the online registration system of the ICZN, which is a requirement in such cases.

Experimental design

The authors have given detailed morphological descriptions of the parts of the trophozoites – epimerite, protomerite, and deutomerite. It is important to note that the coefficient of variation indicates the stability of the morphometric parameters and indices. I agree with the results, which confirm that the almost unchanged characteristics during the growth of gregarines, such as the ratios EL/PECL, PL/PECL, EL/EW, PL/PW, PW/SW, DW/PW, NL/NW and DLA, can be used as reliable and stable diagnostic characteristics in the description of the species. In the discussion, the authors correctly comment on closely related hosts such as Licinus punctulatus and the parasites Actinocephalus and Ramicephalus (Apicomplexa) found in them. The comparative analysis of the morphological and structural differences supports the authors' thesis to distinguish a new species and genus. The details they provide their final conclusions based on sufficient evidence. As a structure of reasoning and comments, I find it important that the authors cite R.E. Clopton (Clopton, 2002; Clopton & Nolte, 2002; Clopton et al., 1992) and adhere to the descriptions and criteria that he uses as a leading specialist in this group.

Validity of the findings

As a remark, I would note the following:
- In the title, it is not clear that they are founding a new genus; it should be written – n. gen., n. sp.
- Fig. 1 – there is no photo with № 9, and such a photo is described in the explanation of the figure.
- I suggest adding a photo of Syzygy, if the authors have one.
- There are no photos of gametocysts and oocysts presented. Why?

Additional comments

Figures 2-5 support the authors' statements and bring clarity and categoricality to the stated thesis. The tables are also informative and logically support the arguments presented by the authors with facts. The explanations are clear and correspond to the terminology used for this group of organisms.

---

## Round 0.2 · Minor Revisions

Please address Reviewer 1's comments about the red line in Fig. 3. You state that this is a very important comment but you do not comment on whether the red trend line should be displayed for all of the plots shown in Fig. 3, or whether some of them (e.g. the one with R(2) = 0.285) should have the red line removed. If it should be removed please remove it and resubmit Fig. 3. That is the only point preventing acceptance of this article.

Reviewer 1 ·

Basic reporting

Most parts of the manuscript have been improved.

Line 32: In formal writing, it should be “does not”.

Line 183: Change to “Across the gregarine specimens.”

Figure 1:Thee authors argue that their article is anyway only for experts. I still think that it does not harm to be transparent and to label at one individual what/where the epimerite, protomerite, etc, are. This would be simply proper scientific practice.

Figs. 3 and 4: Maybe there is a misunderstanding. It is fine and correct to add the formulas and R2-values to the graphs. However, it is not correct to add a red line that indicates a significant correlation if this is not significant, which is probably the case if the R2 is much lower than 0.1 (simply test the correlation for significance!). Thus, the red lines should be removed in all cases where the correlation is not significant, while the formula can always be shown.

Experimental design

Fine.

Validity of the findings

Fine.

Reviewer 2 ·

Basic reporting

-

Experimental design

Adequate

Validity of the findings

The findings are valid.

---

## Round 0.3 · accepted · Accept

The reviewer's concerns have been addressed.

Reviewer 1 ·

Basic reporting

The figures are now nicely improved. The second revision is fully satisfactory.

Experimental design

ok

Validity of the findings

ok

Additional comments

na